# β-sitosterol alleviated HFD-induced atherosclerosis by regulating the MAPK/Nrf2/NLRP3 pathway in ApoE−/− mice

Weiping Wu[1], Wugao Liu[1], Ningjun Wu[1], Chunsheng Qu[1], Weihua Chu[2], Jing Jin[1]*

1 Department of Clinical Laboratory, Lishui People's Hospital, The Sixth Affiliated Hospital of Wenzhou Medical University, Lishui, Zhejiang, China, 2 Department of Microbiology, School of Life Science and Technology, China Pharmaceutical University, Nanjing, China

* 46805633@qq.com

## Abstract

### Background

Atherosclerosis (AS), driven by chronic inflammation and oxidative stress, remains a leading cause of cardiovascular morbidity. While β-sitosterol, a dietary phytosterol, shows therapeutic potential for AS, its mechanisms remain unclear. This study aimed to explore whether β-sitosterol alleviates AS by modulating the MAPK/Nrf2/NLRP3 pathway.

### Methods

ApoE−/− mice fed a high-fat diet (HFD) were treated with β-sitosterol for 8 weeks. Lipid profiles, aortic plaque area, oxidative stress markers, and inflammatory mediators were analyzed. Nrf2 pathway activity and NLRP3 inflammasome components were assessed using ELISA, qRT-PCR, and histochemical assays.

### Results

β-sitosterol significantly reduced serum total cholesterol, LDL-C, and aortic plaque area in HFD-fed mice. It suppressed the MAPK pathway and NLRP3 inflammasome activation while downregulating MMP-2/9 expression. Additionally, β-sitosterol activated the Nrf2 pathway, increasing catalase protein (CAT) activity and reducing oxidative stress in liver tissue. However, it showed limited effects on NF-κB, IL-6, IL-10, and certain antioxidants.

### Conclusion

β-sitosterol ameliorates AS by attenuating lipid accumulation, inflammation, and oxidative stress via coordinated regulation of the MAPK/Nrf2/NLRP3 pathways. These findings highlight its potential as a therapeutic agent, though clinical studies are warranted to confirm efficacy and safety in humans.

**Data availability statement:** All relevant data are within the manuscript.

**Funding:** This research was supported by the Joint Funds of the Zhejiang Provincial Natural Science Foundation of China under Grant No. LLSSY24H020002. The funder had no role in study design, data collection and analysis, decision to publish, or preparation of the manuscript. There was no additional external funding received for this study.

**Competing interests:** The authors have declared that no competing interests exist.

## Introduction

Atherosclerosis (AS) is a chronic inflammatory condition characterized by the accumulation of lipids and inflammatory cells within the arterial intima, leading to the formation of atherosclerotic plaques [1]. These vulnerable plaques (VP) pose a significant risk as their rupture or dislodgement can precipitate acute coronary syndromes (ACS), cerebrovascular accidents, and other life-threatening events, often resulting in severe long-term complications such as physical disability, cognitive decline and depression [2]. Although inflammation, lipid accumulation, oxidative stress, and endothelial injury are widely recognized as key contributors to the disease process, the intricate mechanisms underlying the pathogenesis of AS have not been fully elucidate [3–4]. Consequently, there is an urgent need to develop novel pharmacological agents that can safely and effectively prevent and treat AS.

β-sitosterol, recognized as one of the most abundant dietary phytosterols, is found in significant quantities in various plants and shares a structural resemblance to cholesterol, distinguished by an additional ethyl group at the C-24 position [5]. In humans, this bioactive compound cannot be synthesized internally and must be obtained solely through dietary intake via intestinal absorption. β-sitosterol is widely regarded as a safe and promising nutritional component, with a long history of use in pharmaceutical applications, where it typically exhibits minimal adverse effects [6]. Among its numerous pharmacological properties, β-sitosterol has remarkable anti-inflammatory effects against conditions such as arthritis, diabetes, cardiovascular diseases, liver diseases and various types of cancers [7–11].

Nucleotide-binding oligomerization domain-like receptor protein 3 (NLRP3) inflammasome is activated during inflammation, which plays a crucial role in the pathogenesis of AS [12]. Zhang et al. demonstrated that polydatin effectively inhibits the NLRP3 inflammasome and caspase-1, thereby preventing pyroptosis and the secretion of IL-18 in AS [13]. The inflammatory response is vital in shifting plaques from a stable to an active state, with oxidized low-density lipoprotein (ox-LDL) playing a significant role in this process [14]. This transformation not only facilitates the release of proinflammatory factors that activate the NLRP3 inflammasome but also accelerates cellular pyroptosis [15]. Nuclear factor erythroid 2-related factor 2 (Nrf2), as an important transcription factor, regulates the cellular defense against oxidative stress [16–17]. Nrf2 can bind to the antioxidant response element and regulate the levels of antioxidant enzymes such as heme oxygenase-1 (HO-1) [18]. Yin et al. showed that curcumin can improve the inflammatory state by activating the Nrf2 pathway [19]. Kynurenic acid can inhibit the production of reactive oxygen species (ROS) by activating the Nrf2 pathway [20].

Our previous research found that β-sitosterol has therapeutic potential for treating AS by inhibiting trimethylamine production and inflammatory reaction [21]. However, several issues remain to be solved, including the potential mechanisms underlying the therapeutic effect of β-sitosterol. Therefore, this study attempts to investigate the potential signaling pathways underlying therapeutic mechanisms of β-sitosterol in AS.

## Materials and methods

### Animals and treatments

Six male C57BL/6J mice (6 ± 1 weeks) and 12 Apolipoprotein E knockout mice (ApoE$^{-/-}$ mice) (6 ± 1 weeks) were purchased from Cavins Experimental Animal Technology Co., Ltd. (Changzhou, China) and were kept in standard polypropylene cages at 22 ± 2 °C and 55 ± 5% relative humidity with a 12-h light/dark period. The use of mice in this experiment was approved by the China Pharmaceutical University Animal Care and Use Committee, and the whole process of the animal experiments followed the guidelines of the Institute Animal Care and Use Committee of China Pharmaceutical University (2023YD0059). The sample size of 6 mice per group was based on empirical data and the design of previous similar experiments [13,22,23]. Every effort was made to minimize the number and suffering of mice included in this study.

Con group (n = 6): C57BL/6J mice fed a standard chow diet with 0.1 ml olive oil by gavage. The ApoE$^{-/-}$ mice were randomly divided into 2 groups with 6 mice in each group. AC group: AS model mice given a high-fat diet (HFD) containing 1.25% cholesterol and 40% fat with 0.1 ml olive oil by gavage. AT group: experimental mice given a HFD with 400 mg/kg/d β-sitosterol (Plant Origin Biological, Nanjing, China) by gavage. All groups were given gavage once a day, their water was changed every two days, and their body and food intake were weighed and recorded weekly. Eight weeks later, the mice were anesthetized with 50 mg/kg pentobarbital sodium and euthanized by cervical dislocation, and serum, aortic tissues and liver tissues were collected for subsequent experiments.

### Histopathological and cholesterol metabolism analysis

Atherosclerotic plaques were analyzed in the aortas by Oil Red O (ORO) staining and HE staining. The aortic root was fixed in 4% paraformaldehyde, sectioned into serial 10 μm slices, which were stained with ORO or HE and observed under a microscope scanner (Pannoramic MIDI II, 3DHISTECH, Budapest, Hungary) to obtain pathological image information (the atherosclerotic plaques were stained red). The liver tissues of mice were collected, frozen in liquid nitrogen, sectioned into slices and stained with ROS fluorescence. The plaque area and fluorescence intensity of ROS were quantified by Image J software.

The serum levels of total cholesterol (T-CHO), TG, low-density lipoprotein cholesterol (LDL-C) and high-density lipoprotein cholesterol (HDL-C) were detected according to the manufacturer's instructions (Jiancheng Bioengineering Institute, Nanjing, China).

### Proinflammatory cytokines and antioxidant activity analysis

ELISA kits (Youxuan Biotechnology Co., Ltd. Shanghai, China) were used to detect the inflammation-related cytokines (IL-10, IL-6), tumor necrosis factor (TNF-α), p38 mitogen-activated protein kinase (p38 MAPK), extracellular signal-regulated kinase (ERK), NLRP3, JNK and NF-κB in the aorta and Nrf2, quinone oxidoreductase 1 (NQO1) and CAT in liver tissues. The activity of glutathione peroxidase (GSH-Px), superoxide dismutase (SOD) activities, CAT and the level of malondialdehyde (MDA) in liver tissues were assayed using the respective commercial kits according to the manufacturer's instructions (Jiancheng Bioengineering Institute, Nanjing, China).

Total RNA was extracted from aortic tissues using FreeZol reagent (Vazyme International LLC, Nanjing, China). Matrix metalloproteinases (MMP)-2 and MMP-9 in the aorta were detected by HiScript II One Step qRT-PCR SYBR Green Kit (Vazyme International LLC, Nanjing, China). Primer sequences (Sangon Biotech Co., Ltd. Shanghai, China) were given in Table 1. U6 was used as a reference gene. The specificity of the primers was verified by fusion curve analysis. The expression abundance of related genes was determined by the Ct value, and the relative gene expression was calculated by $2^{-\Delta\Delta Ct}$.

**Table 1. Primers used in this study.**

| Gene | Primer sequence (5'-3') |
| --- | --- |
| MMP-2 | FOR: CCCTCAAGAAGATGCAGAAGTTC<br>REV: TCTTGGCTTCCGCATGGT |
| MMP-9 | FOR: CGTCGTGATCCCCACTTACT<br>REV: AACACACAGGGTTTGCCTTC |
| U6 | FOR: CTCGCTTCGGCAGCACA<br>REV: AACGCTTCACGAATTTGCGT |

## Statistical analysis

All data are presented as the mean ± SD. GraphPad Prism 9.5.1 (GraphPad Software, La Jolla, CA, USA) was used for data analysis. Comparisons between two groups were analyzed using one-way ANOVA. Values of $p < 0.05$, $p < 0.01$, $p < 0.001$ and $p < 0.0001$ indicate significant differences.

## Results

### β-sitosterol alleviated lipid metabolism and aortic root plaque in ApoE⁻/⁻ mice

There was no significant difference in body weight among the groups (Fig 1A). Compared with the Con group, T-CHO, TG, LDL-C and HDL-C in the AC group were significantly increased. After β-sitosterol treatment, T-CHO, LDL-C and HDL-C in the AT group were significantly decreased (Fig 1B-1E). However, β-sitosterol had no significant effect on TG level. To study the effect of β-sitosterol on aortic root plaque, the formation of atherosclerotic plaques was visualized by ORO staining and HE staining. As shown in Fig 1F-1G, HFD consumption significantly increased the aortic root plaques in the AC group, whereas the treatment of β-sitosterol reversed this effect in the AT group.

### β-sitosterol inhibited the MAPK signaling pathway in the aortic roots of ApoE⁻/⁻ mice

As depicted in Fig 2A-2E, HFD feeding significantly increased the levels of NF-κB, p38 MAPK, ERK, NLRP3 and JNK. β-Sitosterol treatment significantly reduced p38 MAPK, ERK, NLRP3 and JNK, but not NF-κB. Moreover, our findings indicated that MMP-2 and MMP-9 were downregulated in the aortic plaques of mice in the AT group compared to those in the AC group (Fig 2F-2G). However, apart from TNF-α, there were no differences in the proinflammatory protein IL-6 and the anti-inflammatory protein IL-10 in the aortic root of mice. These results indicated that β-sitosterol could inhibit the formation of aortic plaques and the NLRP3 inflammasome through MAPK signaling pathway.

### β-sitosterol reduced oxidative stress by activating Nrf2 signaling pathway in liver tissue of ApoE⁻/⁻ mice

As depicted in Fig 3A and 3D, nuclear Nrf2 and CAT were upregulated by β-sitosterol. Similarly, the activity of CAT was significantly reversed by β-sitosterol in the liver tissues of ApoE⁻/⁻ mice (Fig 3F). However, compared with those in the Con group, the activity of GSH-Px and the level of MDA in the AC group were significantly different, and there was no difference between the AC and AT groups (Fig 3G, 3H). In addition, β-Sitosterol had a weaker effect on NQO1, HO-1, SOD activity and intracellular ROS generation (Fig 3B, 3C, 3E, 3I). These findings revealed that β-Sitosterol reduced oxidative stress by activating the Nrf2 signaling pathway in the liver tissues of ApoE⁻/⁻ mice.

## Discussion

Atherosclerosis is a complex and multifactorial disease closely linked to HFD consumption. In this study, we investigated the effect of β-sitosterol on HFD-induced atherosclerosis in ApoE⁻/⁻ mice and explored the underlying mechanisms involving the MAPK/Nrf2/NLRP3 pathway. Our results demonstrated that β-sitosterol significantly alleviated HFD-induced

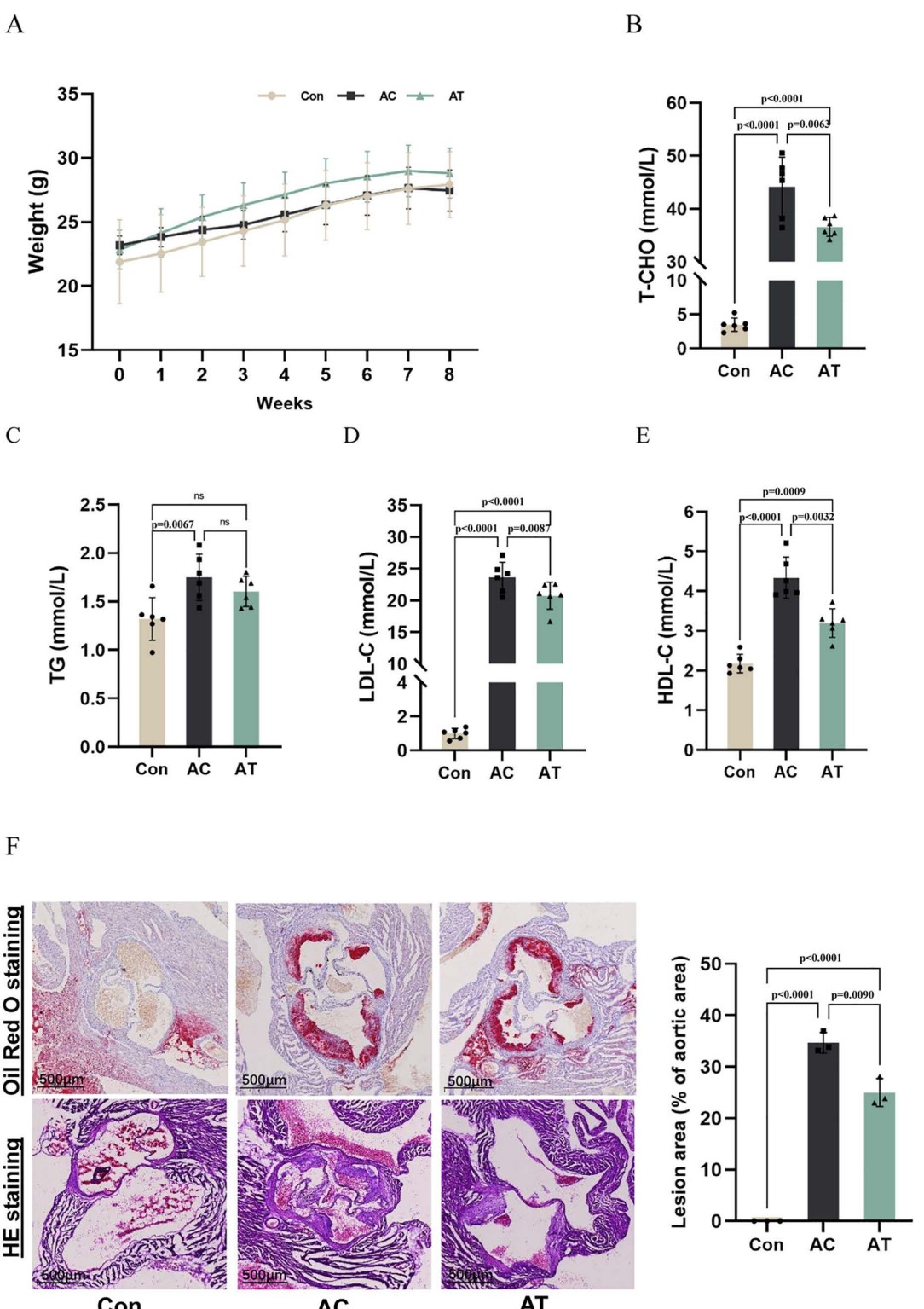

**Fig 1. β-sitosterol alleviated lipid metabolism and aortic root plaque in HFD-fed ApoE$^{-/-}$ mice.** (A) Effect of β-sitosterol on body weight in ApoE$^{-/-}$ mice (n = 6). (B-E) Effect of β-sitosterol on serum T-CHO, TG, LDL-C and HDL-C levels in ApoE$^{-/-}$ mice (n = 6). (F) β-sitosterol reversed the atherosclerotic lesions (n = 3). The aortic root was assessed by Oil Red O staining, and the plaques area were quantified by Image J. Comparisons between two groups were analyzed using one-way ANOVA. ns indicates no significant difference.

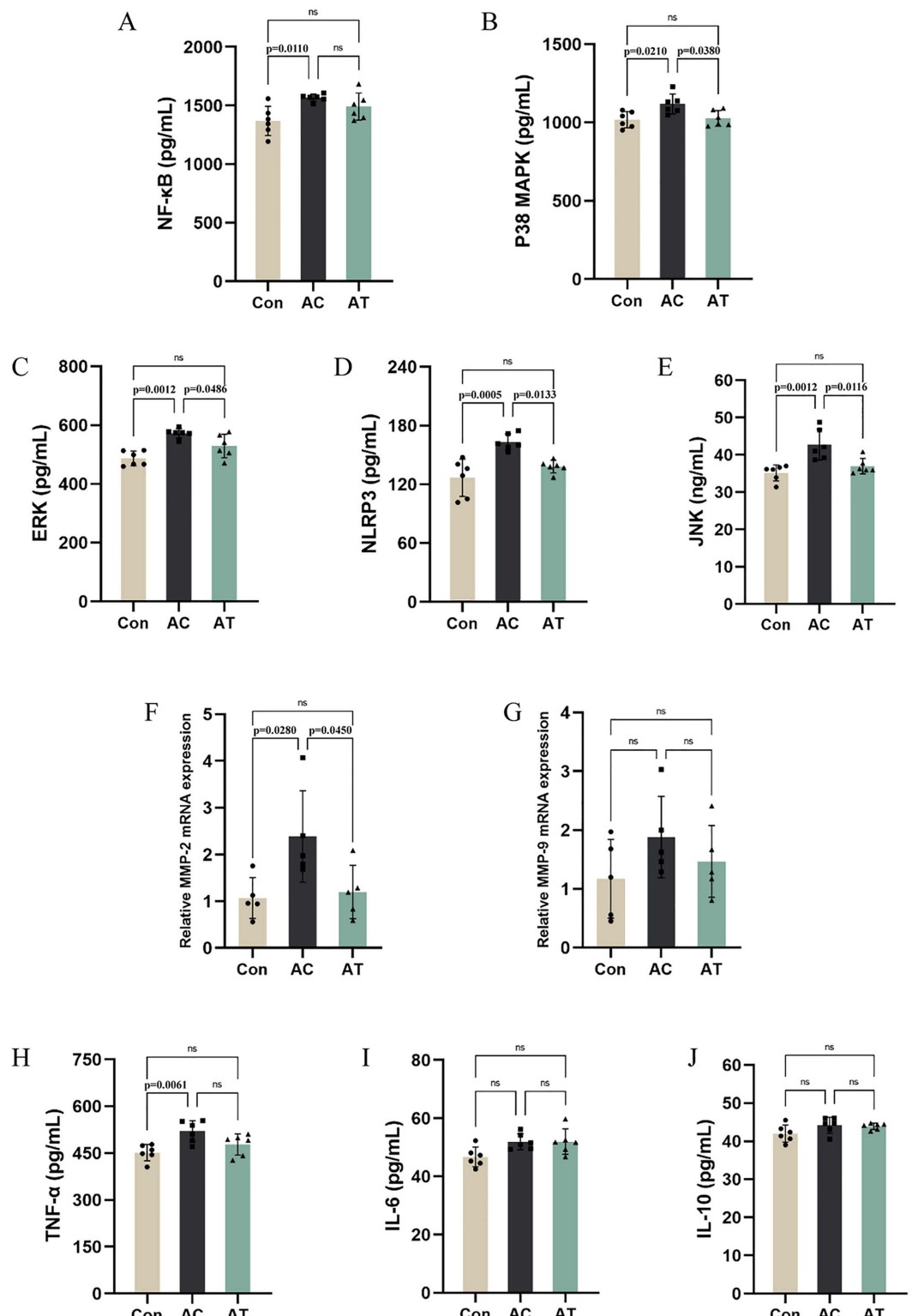

**Fig 2. Effect of β-Sitosterol on MAPK signaling pathway in aorta of HFD-fed ApoE−/− mice (n = 6). (A-E)** ELISA analysis of NF-κB, p38 MAPK, ERK, NLRP3 and JNK. **(F-G)** qRT-PCR analysis of MMP-2 and MMP-9. **(H-J)** ELISA analysis of IL-10, IL-6 and TNF-α. Comparisons between two groups were analyzed using one-way ANOVA. ns indicates no significant difference.

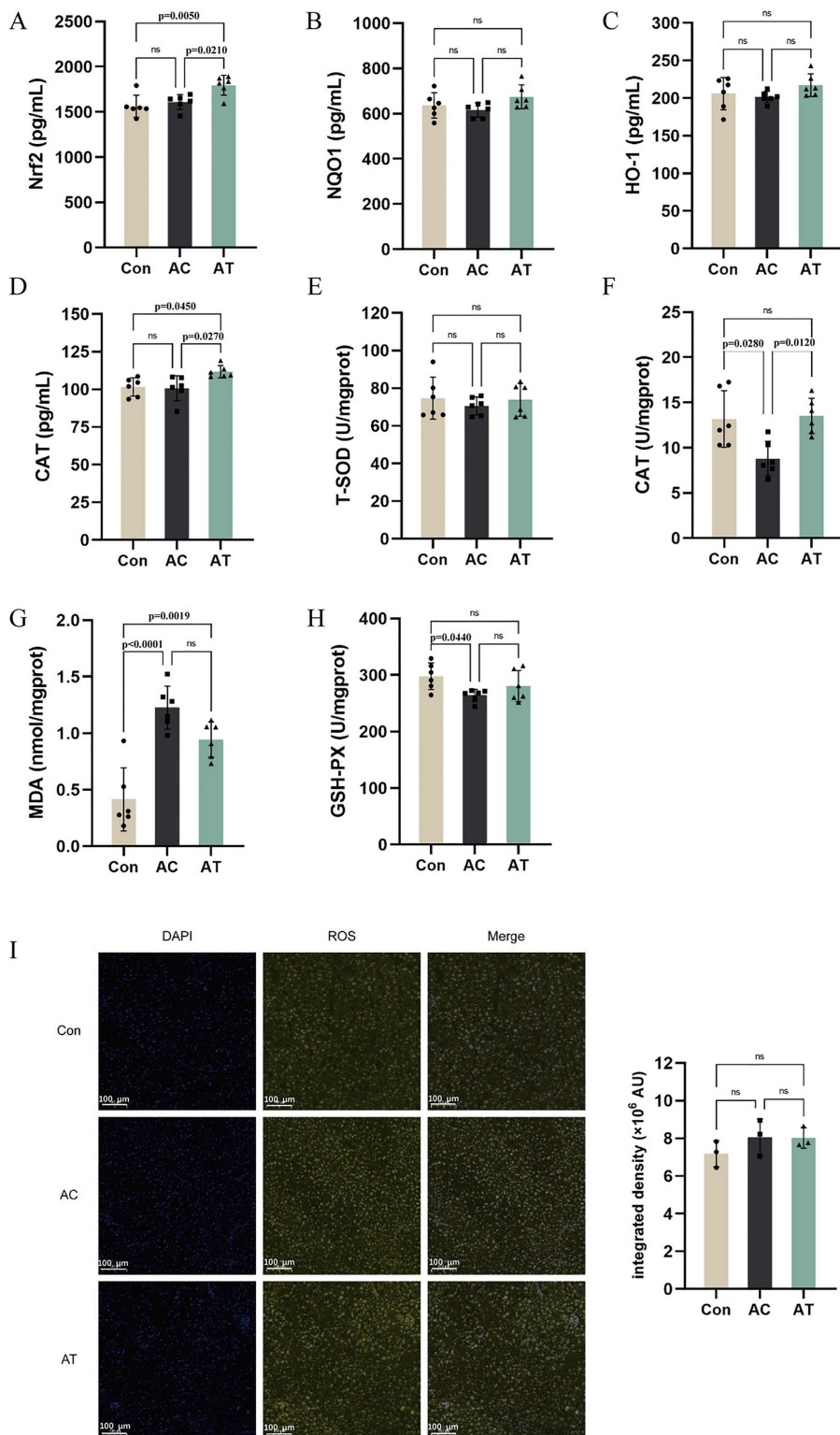

**Fig 3. Effect of β-Sitosterol on Nrf2 signaling pathway in liver tissue of HFD-fed ApoE$^{-/-}$ mice. (A-D)** ELISA analysis of Nrf2, NQO1, HO-1 and CAT (n = 6). **(E-H)** The activity of GSH-Px, SOD and CAT and the level of MDA (n = 6). **(I)** Representative immunofluorescence microscopic images of liver ROS, scale bar = 100 μm (n = 3). Comparisons between two groups were analyzed using one-way ANOVA. *$p < 0.05$, **$p < 0.01$, ***$p < 0.001$, ****$p < 0.0001$, ns indicates no significant difference.

atherosclerosis in ApoE$^{-/-}$ mice. This was evidenced by reduced lipid deposition and a decreased plaque area, as observed through histological staining. These findings are consistent with previous studies suggesting that β-sitosterol has anti-atherosclerotic properties. For instance, β-sitosterol inhibited plaque formation and platelet activation, and decreased serum TC and TG levels [24]. MMPs promote extracellular matrix (ECM) degradation and weaken the fibrous caps of plaques, ultimately promoting plaque rupture [25]. MMPs play pivotal roles in the occurrence, development, and complication formation of AS. In the early stage of AS, MMPs are involved in macrophage migration [26]. As AS progresses, MMPs not only degrade the ECM but also affect the function of smooth muscle cells [27]. In the late stage of AS, the overexpression of MMPs is one of the key factors leading to plaque rupture [28]. Consistent with these studies, MMP2 and MMP9 levels were elevated in the AC group but were reduced to normal levels by β-sitosterol treatment in our study, indicating the protection effect of β-sitosterol on AS.

The MAPK signaling pathway, which includes ERK, JNK, and p38, plays a pivotal role in regulating various cellular responses, including inflammation and cell proliferation, which are key processes in atherosclerosis development [29–30]. In our study, HFD-induced activation of the MAPK pathway was evidenced by increased levels of ERK, JNK, and p38. β-sitosterol effectively suppressed the activation of the MAPK pathway. This is in line with research on other natural compounds. For example, curcumin, a polyphenol, was found to inhibit the MAPK pathway, thereby reducing inflammation in a colitis model [17]. Nrf2 is a critical transcription factor in the cellular defense against oxidative stress [31]. Under normal conditions, Nrf2 is sequestered in the cytoplasm by Keap1. However, under oxidative stress, Nrf2 dissociates from Keap1, translocates to the nucleus, and binds to antioxidant response elements, promoting the expression of antioxidant genes like HO-1 and NQO1 [32–33]. In our study, HFD consumption downregulated Nrf2, resulting in a decrease in the expression of antioxidant enzymes such as CAT, while β-sitosterol treatment reversed this effect. These findings indicate that β-sitosterol enhances the antioxidant capacity of cells, protecting them from oxidative damage. Similar findings were reported for resveratrol, which activated the Nrf2 pathway and alleviated oxidative stress in a cardiovascular disease model [34]. The activation of the Nrf2 pathway by β-sitosterol may be a key mechanism involved in reducing the oxidative stress that drives atherosclerosis progression.

The NLRP3 inflammasome is a multiprotein complex that plays a central role in the inflammatory response. The activation of the NLRP3 inflammasome leads to the cleavage of procaspase-1 into active caspase-1, which in turn promotes the maturation and secretion of proinflammatory cytokines that contribute to atherosclerosis [35]. Some studies have identified the important role of NLRP3 activation in exacerbating oxidative stress [36–37]. On the other hand, NLRP3 is also activated by oxidative stress, with elevated levels of ROS serving as a prominent trigger [38]. Our results showed that HFD upregulated NLRP3 expression, whereas β-sitosterol treatment effectively inhibited this upregulation. This is consistent with studies on other natural products, such as quercetin, which has been shown to suppress NLRP3 inflammasome activation and reduce inflammation in atherosclerotic lesions [39]. By inhibiting the NLRP3 inflammasome, β-sitosterol may attenuate the chronic inflammatory state and oxidative stress in atherosclerotic plaques.

There is a complex interplay among the MAPK, Nrf2, and NLRP3 pathways. Activation of the MAPK pathway can lead to increased oxidative stress, which in turn can activate the NLRP3 inflammasome. Conversely, activation of the Nrf2 pathway can counteract oxidative stress and inhibit NLRP3 inflammasome activation. β-sitosterol appears to modulate these pathways in a coordinated manner. By suppressing the MAPK pathway, β-sitosterol reduces the inflammatory state, which may indirectly inhibit NLRP3 inflammasome activation. Simultaneously, the activation of the Nrf2 pathway by β-sitosterol further contributes to the inhibition of NLRP3 inflammasome activation through its antioxidant effects. This coordinated regulation of multiple pathways likely underlies the anti-atherosclerotic effects of β-sitosterol. Besides, while our data indicated that β-sitosterol did not significantly alter NF-κB levels, this finding aligns with prior studies demonstrating that β-sitosterol's anti-inflammatory effects may be context-dependent and pathway-specific. For instance, β-sitosterol has been shown to suppress inflammation primarily through MAPK/Nrf2/NLRP3 modulation rather than direct NF-κB inhibition [40].

Our study has some limitations. First, these findings primarily rely on expression data alone, the exact molecular mechanisms by which β-sitosterol interacts with these pathways remain to be fully elucidated. Future studies could use more advanced techniques, such as gene knockout or overexpression models, to further explore these mechanisms. Second, the study was conducted in ApoE$^{-/-}$ mice, and the results may not be directly applicable to humans. Clinical trials are needed to confirm the efficacy and safety of β-sitosterol in human atherosclerosis patients.

In conclusion, our study demonstrated that β-sitosterol alleviated HFD-induced atherosclerosis in ApoE$^{-/-}$ mice by regulating the MAPK/Nrf2/NLRP3 pathway. These findings suggest that β-sitosterol has potential as a therapeutic agent for atherosclerosis. However, further research is required to fully understand its mechanisms of action and to evaluate its efficacy in humans.

## Author contributions

**Conceptualization:** Jing Jin.

**Data curation:** Weiping Wu, Jing Jin.

**Formal analysis:** Jing Jin.

**Funding acquisition:** Jing Jin.

**Investigation:** Weiping Wu.

**Methodology:** Weiping Wu, Ningjun Wu.

**Project administration:** Weiping Wu, Wugao Liu, Ningjun Wu.

**Supervision:** Ningjun Wu.

**Visualization:** Wugao Liu.

**Writing – original draft:** Weiping Wu, Wugao Liu.

**Writing – review & editing:** Ningjun Wu, Chunsheng Qu, Weihua Chu.

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
