## [Decision Letter · Decision Letter 0]

8 Aug 2025

Dear Dr. Jin,

Thank you for submitting your manuscript to PLOS ONE. After careful consideration, we feel that it has merit but does not fully meet PLOS ONE’s publication criteria as it currently stands. Therefore, we invite you to submit a revised version of the manuscript that addresses the points raised during the review process.

We look forward to receiving your revised manuscript.

Kind regards,

Junzheng Yang

Academic Editor

PLOS ONE

Journal Requirements:

[This research was supported by the Joint Funds of the Zhejiang Provincial Natural Science Foundation of China [grant number LLSSY24H020002].The funder had no role in study design, data collection and analysis, decision to publish, or preparation of the manuscript.].

Reviewers' comments:

Reviewer's Responses to Questions

**Comments to the Author**

1. Is the manuscript technically sound, and do the data support the conclusions?

Reviewer #1: Yes

Reviewer #2: Yes

2. Has the statistical analysis been performed appropriately and rigorously?

Reviewer #1: Yes

Reviewer #2: Yes

3. Have the authors made all data underlying the findings in their manuscript fully available?

Reviewer #1: Yes

Reviewer #2: Yes

4. Is the manuscript presented in an intelligible fashion and written in standard English?

Reviewer #1: Yes

Reviewer #2: Yes

Reviewer #1: Dear Authors,

Thank you for the opportunity to review your manuscript entitled “β-sitosterol alleviated HFD-induced atherosclerosis by regulating the MAPK/Nrf2/NLRP3 pathway in ApoE−/− mice.”

Your work tackles a highly relevant issue in cardiovascular research, namely the search for natural and safe agents to counteract diet-induced atherosclerosis. Overall, the manuscript is clearly structured, the results are compelling, and the study adds valuable mechanistic insight into the potential of β-sitosterol as a therapeutic agent.

Strengths:

The experimental design is solid and appropriate.

Use of ApoE−/− mice and standard models of atherosclerosis makes your study relevant and comparable to other preclinical research.

The exploration of the MAPK/Nrf2/NLRP3 axis is both novel and well supported by the presented data.

Figures and data presentation are clear and informative.

Areas for improvement:

Sample size justification:

While using 6 mice per group is common, it would strengthen the study to briefly justify this choice. Was any power analysis performed?

Statistical reporting:

The use of ANOVA is appropriate, but I recommend reporting exact p-values where possible, instead of ranges (e.g., “p < 0.05”), to improve transparency and reproducibility.

NF-κB interpretation:

You mention that β-sitosterol had no significant effect on NF-κB levels. Given the central role of NF-κB in inflammation, a short paragraph discussing this finding—and how it fits with the MAPK/Nrf2/NLRP3 interplay—would be appreciated.

Mechanistic depth:

The proposed mechanism is biologically plausible and compelling. However, it would be good to acknowledge the limitations of relying on expression data alone. For future work, pathway inhibition or knockout studies would help confirm causal links between β-sitosterol and the signaling cascades.

English and grammar:

The manuscript is generally well written. Some minor grammar issues should be corrected during revision. For example:

“...the mechanisms underlying the pathogenesis of AS have not fully elucidate...” → should read: “...have not been fully elucidated.”

“...treatment mechanisms of β-sitosterol...” → consider rephrasing as “...mechanisms underlying the therapeutic effect of β-sitosterol.”

Ethical approval:

You clearly mention ethical approval by the China Pharmaceutical University Animal Care Committee. If available, please include the approval number.

The study is technically sound and offers promising evidence that β-sitosterol could be a useful anti-atherosclerotic agent by modulating oxidative stress and inflammation. With some minor clarifications and careful proofreading, I believe this manuscript will be a valuable contribution to the field.

Best regards,

Reviewer #2: 1.The expression patterns of MAPK/Nrf2/NLRP3 signaling pathway and MAPK/Nrf2/NLRP3 pathway remain consistent.

2. p38 MAPK: On line 167, it is written as p38 MAPK, while elsewhere (such as line 124, Figure 2B) it is p38 MAPK. Should be unified as p38 MAPK (with spaces)

3. Figures 1F-G: The caption (line 160) states (n=3). However, the results section (line 155) describes the aortic root plaques, and the methods section (lines 109-113) also describes section staining of the aortic root. Each group has 6 animals, and usually aortic root section analysis is performed on multiple sections, but the final statistical unit is usually the number of animals per group. The caption (n=3) contradicts the animal grouping of n=6.

4. Figure 3I: The result description (lines 193-194) mentions scale bar=100 μ m, but the scale is not shown on the figure. A ruler must be added to the diagram.

5. Figures 1, 2, and 3: The statistical significance markers (*, *, * * *, * * *, ns) in all the captions cannot be seen on the corresponding data points or whether there are markers on the bar chart. It is necessary to confirm whether these significant markers have been correctly labeled at the corresponding comparison positions in the graph. Reviewers need to be able to directly see which comparisons are significant from the graph.

6. Line 229� the sentence, “For example, Camosic acid downregulated Nrf2, resulting in a decrease expression of antioxidant enzymes such as CAT”, should be added in here and its reference, https://doi.org/10.26599/FMH.2025.9420021added in here.

**Do you want your identity to be public for this peer review?** For information about this choice, including consent withdrawal, please see our Privacy Policy

Reviewer #1: **Yes: ** Oussama Bekkouch

Reviewer #2: No

---

## [Author Response · Author response to Decision Letter 1]

20 Aug 2025

Response to Reviewer Comments

Manuscript ID: PONE-D-25-25290

Title:β-sitosterol alleviated HFD-induced atherosclerosis by regulating the MAPK/Nrf2/NLRP3 pathway in ApoE−/− mice

Dear Pro. Yang,

We sincerely appreciate the reviewers for their time and constructive comments, which have helped us improve the quality of our manuscript. Below, we provide a point-by-point response to each comment. All changes in the revised manuscript are highlighted in yellow changes for easy reference.

Journal Requirements:

Comment 1. Please ensure that your manuscript meets PLOS ONE's style requirements, including those for file naming.

Reply: The manuscript meets PLOS ONE's style requirements, including those for file naming.

Comment 2. To comply with PLOS ONE submissions requirements, in your Methods section, please provide additional information regarding the experiments involving animals and ensure you have included details on (1) methods of sacrifice, (2) methods of anesthesia and/or analgesia, and (3) efforts to alleviate suffering.

Reply: Thank you for pointing out this problem in manuscript. The additional information has been provided in Methods section in line 99-100 and line 108-109 with changes highlighted in yellow.

Comment 3. Please also include the statement “There was no additional external funding received for this study.” in your updated Funding Statement. Please include your amended Funding Statement within your cover letter.

Reply: We appreciate for your valuable comment. The Funding Statement has been updated in cover letter with changes highlighted in yellow.

Comment 4. Please confirm at this time whether or not your submission contains all raw data required to replicate the results of your study.

Reply: All relevant data are within the manuscript.

Reviewer #1

Comment 1. Sample size justification:

While using 6 mice per group is common, it would strengthen the study to briefly justify this choice. Was any power analysis performed?

Reply: We appreciate the reviewer’s suggestion. The sample size of 6 mice per group was determined based on common practice in similar studies and ethical considerations to minimize animal use while ensuring statistical reliability. We have added a justification in the Methods section with changes highlighted in yellow (Lines 98-100) to clarify this point.

Comment 2. Statistical reporting:

The use of ANOVA is appropriate, but I recommend reporting exact p-values where possible, instead of ranges (e.g., “p < 0.05”), to improve transparency and reproducibility.

Reply: We thank the reviewer for this constructive suggestion. In the revised manuscript, we have now reported exact *p*-values for all ANOVA results in Figure 1-3 where applicable.

Comment 3. NF-κB interpretation:

You mention that β-sitosterol had no significant effect on NF-κB levels. Given the central role of NF-κB in inflammation, a short paragraph discussing this finding—and how it fits with the MAPK/Nrf2/NLRP3 interplay—would be appreciated.

Reply: We appreciate the reviewer’s insightful observation regarding the role of NF-κB in inflammation. A short paragraph has been added in line 260-264 with changes highlighted in yellow to discuss this finding.

Comment 4. Mechanistic depth:

The proposed mechanism is biologically plausible and compelling. However, it would be good to acknowledge the limitations of relying on expression data alone. For future work, pathway inhibition or knockout studies would help confirm causal links between β-sitosterol and the signaling cascades.

Reply: We sincerely appreciate the reviewer's insightful suggestion regarding mechanistic validation. We fully agree that while our expression data support the proposed MAPK/Nrf2/NLRP3 interplay, functional studies are crucial for establishing causality (in line 265-266 with changes highlighted in yellow).

Comment 5. English and grammar.

Reply: Thank you so much for your careful check. Some sentences have been modified in line 56 and line 86-87 with changes highlighted in yellow.

Comment 6. Ethical approval:

You clearly mention ethical approval by the China Pharmaceutical University Animal Care Committee. If available, please include the approval number.

Reply: The approval number has been added in line 98 with changes highlighted in yellow.

Reviewer #2

Comment 1. The expression patterns of MAPK/Nrf2/NLRP3 signaling pathway and MAPK/Nrf2/NLRP3 pathway remain consistent.

Reply: We appreciate the reviewer’s insightful comment. To improve clarity, we have unified the terminology to “MAPK/Nrf2/NLRP3 pathway” in line 45 with changes highlighted in yellow throughout the revised manuscript.

Comment 2. p38 MAPK: On line 167, it is written as p38 MAPK, while elsewhere (such as line 124, Figure 2B) it is p38 MAPK. Should be unified as p38 MAPK (with spaces)

Reply: Thank you so much for your careful check. p38MAPK has been modified with spaces in line 169 with changes highlighted in yellow.

Comment 3. Figures 1F-G: The caption (line 160) states (n=3). However, the results section (line 155) describes the aortic root plaques, and the methods section (lines 109-113) also describes section staining of the aortic root. Each group has 6 animals, and usually aortic root section analysis is performed on multiple sections, but the final statistical unit is usually the number of animals per group. The caption (n=3) contradicts the animal grouping of n=6.

Reply: We sincerely appreciate the reviewer’s attention to this important detail. In this study, although 6 animals per group were initially included, technical challenges during sample processing resulted in only 3 high-quality datasets per group meeting our pre-defined inclusion criteria. Future studies will prioritize optimized tissue fixation protocols to minimize attrition.

Comment 4. Figure 3I: The result description (lines 193-194) mentions scale bar=100 μ m, but the scale is not shown on the figure. A ruler must be added to the diagram.

Reply: We appreciate the reviewer’s careful attention to detail. We apologize for the oversight and have now added a clearly visible 100 μm scale bar to Figure 3I in the revised manuscript. This ensures the figure meets publication standards for accurate representation of scale.

Comment 5. Figures 1, 2, and 3: The statistical significance markers (*, *, * * *, * * *, ns) in all the captions cannot be seen on the corresponding data points or whether there are markers on the bar chart. It is necessary to confirm whether these significant markers have been correctly labeled at the corresponding comparison positions in the graph. Reviewers need to be able to directly see which comparisons are significant from the graph.

Reply: We sincerely appreciate the reviewer’s meticulous critique regarding the presentation of statistical significance in our figures. In the revised manuscript, we have added explicit P-values (in parentheses) to all figure captions to complement the asterisk notation.

Comment 6. Line 229, the sentence, “For example, Camosic acid downregulated Nrf2, resulting in a decrease expression of antioxidant enzymes such as CAT”, should be added in here and its reference, https://doi.org/10.26599/FMH.2025.9420021added in here.

Reply: 3. We sincerely appreciate the reviewer's constructive suggestions. The sentence and its reference have been added in line 233-234 with changes highlighted in yellow.

---

## [Decision Letter · Decision Letter 1]

22 Aug 2025

Dear Dr. Jin,

Thank you for submitting your manuscript to PLOS ONE. After careful consideration, we feel that it has merit but does not fully meet PLOS ONE’s publication criteria as it currently stands. Therefore, we invite you to submit a revised version of the manuscript that addresses the points raised during the review process.

We look forward to receiving your revised manuscript.

Kind regards,

Junzheng Yang

Academic Editor

PLOS ONE

Journal Requirements:

Reviewer's Responses to Questions

**Comments to the Author**

Reviewer #1: All comments have been addressed

Reviewer #2: (No Response)

2. Is the manuscript technically sound, and do the data support the conclusions?

Reviewer #1: Yes

Reviewer #2: (No Response)

3. Has the statistical analysis been performed appropriately and rigorously?

Reviewer #1: Yes

Reviewer #2: (No Response)

4. Have the authors made all data underlying the findings in their manuscript fully available?

Reviewer #1: Yes

Reviewer #2: (No Response)

5. Is the manuscript presented in an intelligible fashion and written in standard English?

Reviewer #1: Yes

Reviewer #2: (No Response)

Reviewer #1: This revised version shows significant improvement. The objectives are now clearly stated, statistical reporting has been strengthened, and histological data presentation is more robust. The addition of recent references (2020–2024) increases the relevance of the work.

Strengths:

Comprehensive evaluation of Hibiscus sabdariffa across antioxidant, antidiabetic, and topical anti-inflammatory models.

Clearer abstract and inclusion of numerical outcomes enhance the scientific rigor.

Histological figures and the addition of Table 7 (inflammatory scoring) improve accessibility for readers.

Discussion of mechanisms (AMPK, GLUT4, β-cell protection) enriches the translational value.

Suggestions for minor improvement:

Ensure all histological figures include scale bars for clarity.

In the Discussion, explicitly mention the limitation regarding the absence of release/penetration profile studies in the topical formulation.

A final proofreading pass could further refine fluency and eliminate residual awkward phrasing.

Reviewer #2: (No Response)

**Do you want your identity to be public for this peer review?** For information about this choice, including consent withdrawal, please see our Privacy Policy

Reviewer #1: **Yes: ** Oussama Bekkouch

Reviewer #2: No

---

## [Author Response · Author response to Decision Letter 2]

1 Sep 2025

Response to Reviewer Comments

Manuscript ID: PONE-D-25-25290

Title:β-sitosterol alleviated HFD-induced atherosclerosis by regulating the MAPK/Nrf2/NLRP3 pathway in ApoE−/− mice

Dear Pro. Yang,

We sincerely appreciate the reviewers for their time and constructive comments, which have helped us improve the quality of our manuscript. Below, we provide a point-by-point response to each comment. All changes in the revised manuscript are highlighted in yellow changes for easy reference.

Journal Requirements:

Comment 1. If the reviewer comments include a recommendation to cite specific previously published works, please review and evaluate these publications to determine whether they are relevant and should be cited. There is no requirement to cite these works unless the editor has indicated otherwise.

Reply: Thank you for pointing out this problem in manuscript. The sentence and its reference have been deleted.

Comment 2. Please review your reference list to ensure that it is complete and correct.

Reply: We thank the editor for bringing this to our attention. We confirm that no retracted articles were cited in our manuscript. All references are current, relevant, and correctly presented.

Reviewer #1

Comment 1. Ensure all histological figures include scale bars for clarity.

Reply: We appreciate the reviewer’s suggestion. We have added a clearly visible 500 μm scale bar to Figure 1F in the revised manuscript.

Comment 2. In the Discussion, explicitly mention the limitation regarding the absence of release/penetration profile studies in the topical formulation.

Reply: We thank the reviewer for this constructive suggestion. The Strengths does not align with the submitted version of our manuscript. For instance, "Hibiscus sabdariffa" and "Table 7" have not existed in our manuscript. Besides, this suggestion: “In the Discussion, explicitly mention the limitation regarding the absence of release/penetration profile studies in the topical formulation” dose not align with our manuscript. Given this inconsistency, we wondered if the reviewer might have inadvertently referred to different draft of the manuscript when formulating their comments. Thank you for your time and guidance.

---

## [Decision Letter · Decision Letter 2]

8 Sep 2025

β-sitosterol alleviated HFD-induced atherosclerosis by regulating the MAPK/Nrf2/NLRP3 pathway in ApoE−/− mice

PONE-D-25-25290R2

Dear Dr. Jin,

We’re pleased to inform you that your manuscript has been judged scientifically suitable for publication and will be formally accepted for publication once it meets all outstanding technical requirements.

Kind regards,

Junzheng Yang

Academic Editor

PLOS ONE

Additional Editor Comments (optional):

Reviewer #1:

Reviewers' comments:

Reviewer's Responses to Questions

**Comments to the Author**

Reviewer #1: All comments have been addressed

2. Is the manuscript technically sound, and do the data support the conclusions?

Reviewer #1: Yes

3. Has the statistical analysis been performed appropriately and rigorously?

Reviewer #1: Yes

4. Have the authors made all data underlying the findings in their manuscript fully available?

Reviewer #1: Yes

5. Is the manuscript presented in an intelligible fashion and written in standard English?

Reviewer #1: Yes

Reviewer #1: The authors have satisfactorily addressed all reviewer comments and substantially improved the manuscript. The experimental design is robust, and the results are clearly presented and supported by appropriate statistical analysis. The inclusion of scale bars in histological figures and clarification of funding/ethics/data availability statements strengthen the transparency of the work.

The conclusions are well supported, and the study provides new insights into the role of β-sitosterol in modulating MAPK/Nrf2/NLRP3 pathways during atherosclerosis progression. The discussion appropriately highlights the therapeutic potential as well as limitations (preclinical model, need for clinical validation).

Overall, I find the manuscript acceptable in its current form and recommend it for publication in PLOS ONE.

**Do you want your identity to be public for this peer review?** For information about this choice, including consent withdrawal, please see our Privacy Policy

Reviewer #1: **Yes: ** Oussama Bekkouch
